# Improvement of Seed Germination under Salt Stress via Overexpressing *Caffeic Acid O-methyltransferase 1* (*SlCOMT1*) in *Solanum lycopersicum* L.

**DOI:** 10.3390/ijms24010734

**Published:** 2023-01-01

**Authors:** Lianjing Ge, Xiaoyu Yang, Yue Liu, Huimeng Tang, Qifang Wang, Shunpeng Chu, Jinxiang Hu, Ning Zhang, Qinghua Shi

**Affiliations:** College of Horticultural Science and Engineering, Shandong Agricultural University, Tai’an 271018, China

**Keywords:** tomato, *SlCOMT1*, seed germination, salt stress, starch metabolism, osmotic adjustment, antioxidant capacity

## Abstract

Melatonin (MT) is a phytohormone-like substance and is profoundly involved in modulating nearly all aspects of plant development and acclimation to environmental stressors. However, there remain no studies about the effects of MT on tomato seed germination under salt stress. Here we reported that the overexpression of *caffeic acid O-methyltransferase 1* (*SlCOMT1*) significantly increased both MT content and salt tolerance in the germinated seeds of a transgenic tomato relative to wild type (WT) samples. Physiological investigation showed higher amylase activity in the stressed overexpression seeds than WT, leading to the promoted starch decomposition and enhanced soluble sugar content. The stimulated production of osmolytes and enhanced activities of SOD, POD, and CAT, together with the significant reduction in H_2_O_2_ and O_2_^·−^ accumulation, were revealed in the stressed overexpression seeds relative to WT, largely accounting for their lower membrane lipid peroxidation. qPCR assay showed that, upon salt stress, the transcript abundance of hub genes related to germination (*SlCYP707A1*, *SlABA1*, *SlGA3ox2* and *SlGA2ox4*) and stress tolerance (*SlCDPK1*, *SlWRKY33* and *SlMAPK1*) were distinctly altered in the overexpression samples when compared to WT, providing a molecular basis for MT-mediated improvement of seed salt tolerance. Altogether, our observations shed new insights into biological functions of *SlCOMT1* and could expand its utilization in genetic improvement of tomato salt tolerance in future.

## 1. Introduction

Plant seed, the most important reproductive organ, consists of seed coat, cotyledon/endosperm, and embryo, among which the embryo is usually enclosed by the seed coat and the cotyledon/endosperm, and eventually develops into an offspring plant [1]. Upon favorable growth conditions, seed germination can be initiated, which represents the end of stationary state and the resumption of metabolic activities [1,2]. This process involves a series of physiological and biochemical events that are tightly regulated by internal signals, such as nutrients and phytohormones, as well as ambient stimulus/stressors such as light, temperature, and salt stress [3]. Therefore, seed germination is considered the most critical developmental stage in the plant lifecycle, and could impose a significant impact on final product yield and quality, especially under stressful conditions [4].

Salinization is a major soil degradation process in which water-soluble salts, predominantly including Na^+^ and Cl^−^, are excessively accumulated in the soil [5]. It results from a multitude of natural and anthropogenic causes, such as climate changes, poor irrigation management, and improper usage of fertilizers, and has occurred in about 800 million hectares of irrigated field and 32 million hectares of dryland over the past decades [6]. Salt stress now becomes a severe environmental challenge worldwide, greatly constraining the production of both field and horticultural crops [7]. The detrimental effects of salt stress on crop growth and development are mainly derived from osmotic stress at the early stage of salt exposure, which can induce physiological drought and lower carbon assimilation, followed by the overaccumulation of toxic Na^+^ in plants, which can lead to tissue necrosis and early senescence [8]. Regarding the germination process, excessive salt accumulation can, on one hand, compromise water uptake that is required for the reactivation of quiescent seed metabolism, and on the other sand, cause severe damages to seed tissues, particularly the embryo [9]. To cope with the salt stress-derived injuries, plants have developed sophisticated resistance-related networks, including the salt overly sensitive (SOS) pathway, the antioxidant system, multiple phytohormone signalings, and so on [10].

Melatonin (N-acetyl-5-methoxytryptamine, MT) is a newly-discovered phytohormone-like substance distributed broadly in plant organs [11]. Its biosynthesis is initiated from tryptophan and is composed of four consecutive reactions catalyzed by at least six enzymes, including tryptophan hydroxylase (TPH), tryptophan decarboxylase (TDC), tryptamine 5-hydroxylase (T5H), serotonin N-acetyltransferase (SNAT), N-acetylserotonin methyltransferase (ASMT), and caffeic acid O-methyltransferase (COMT) [12,13]. Among these enzymes, COMT is directly associated with the rate-limiting step, and is thus considered a crucial modulator for MT biosynthesis in plants [14]. In the past several years, growing evidence has pointed to the functional significance of MT in multiple physiological and biochemical processes related to seed germination, root initiation, photosynthesis, carbon fixation, plant survival against environmental stressors, etc. [15,16]. For example, Yin et al. have reported that, under salt stress, exogenous application of MT on tomato seedlings can activate core enzymes in the ascorbate-glutathione (ASA-GSH) cycle to scavenge excessive reactive oxygen species (ROS), leading to the enhanced salt tolerance [17]. Chen et al. have found that exogenous MT treatment can promote seed germination by modulating the expression of phytohormone signaling genes, thus resulting in the varied metabolic profiles and the reduced damages to cotton seeds under salt stress [18]. Consistently, the apparent improvement in salt tolerance has been observed in a variety of crop seedlings overexpressing *COMT1*, mainly due to the significantly increased osmotic regulatory and antioxidant capacity in these transgenic plants under salt stress [19,20,21,22]. For example, Liu et al. have reported that the overexpression of *SlCOMT1* can significantly improve the resistance of tomato plants to salt challenge by both stimulating proline production and inhibiting the extraordinary accumulation of toxic hydrogen peroxide (H_2_O_2_) and superoxide (O_2_^·−^) [20]. Intriguingly, while the positive regulatory roles of MT are well documented in plant acclimation to developmental or environmental cues, its opposite or negligible effects have been reported in previous studies. For example, Lv et al. have found that *Arabidopsis* seed germination is not significantly influenced when being exogenously treated with a low MT level, but is substantially restricted with the increase of the exogenous MT level due to its synergistically inhibitory effects on seed germination with ABA signaling [23]. This discrepancy promoted us to further explore MT-mediated crop acclimation to salt stress at different developmental stages.

Tomato is one of the best cash vegetable crops all over the world, and China is ranked as the top country in not only the cultivation area but also the annual yield [24]. Upon salt stress, a series of abnormal developmental events, such as seed germination inhibition, growth delaying, early senescence, and/or even plant death, can be observed in tomato, severely restricting its year-round production [25]. We previously reported that tomato plants overexpressing *SlCOMT1* accumulate more MT, and display an improved acclimation capacity relative to WT plants when being challenged by salt stress at seedling stage [26,27]. However, it had never been explored whether *SlCOMT1* overexpression could benefit tomato seed performance in response to salt challenge or not.

In the present study, a physio-molecular investigation was carried out to unveil the effects of *SlCOMT1* overexpression on tomato seed performance under 75 mM NaCl stress. The apparent increase in three seed germination-related parameters was observed for the overexpression lines in comparison to WT, most likely due to their significant enhancement in starch metabolism, as well as osmotic adjustment and antioxidant capacity. Furthermore, *SlCOMT1* overexpression dramatically increased the RNA abundance of hub genes related to seed germination (*SlCYP707A1*, *SlABA1*, *SlGA3ox2* and *SlGA2ox4*) and stress tolerance (*SlCDPK1*, *SlWRKY33* and *SlMAPK1*), providing molecular explanation for MT-mediated improvement of seed salt tolerance to some extent. Collectively, this study expands our knowledge of *SlCOMT1* functions and could benefit *COMT1*-based genetic engineering for salt tolerance improvement of tomato in future.

## 2. Results

### 2.1. SlCOMT1 Ooverexpression Improves Tomato Seed Germination under Salt Stress

To unveil how endogenous MT influenced seed salt tolerance, the germination performance of two tomato *SlCOMT1* overexpression lines (OE1 and OE2, Figure 1A) was investigated under normal and salt stress conditions. We first evaluated *SlCOMT1* abundance in the germinated seeds of WT and two transgenic lines, and found that, indeed, the expression level of *SlCOMT1* was significantly higher in both OE1 and OE2 samples than WT under normal growth conditions, further conforming these overexpression events at the transcriptional level (Figure 1B). Upon 7-d salt stress, while *SlCOMT1* abundance was upregulated in the germinated seeds of WT and the two overexpression lines, the extents of the expression enhancement were quite distinct: for WT samples, the relative abundance of *SlCOMT1* was increased from 1.00 to 4.92; in contrast, the relative abundance of *SlCOMT1* was increased from 2.56 and 3.30 to 15.12 and 15.62 for OE1 and OE2 samples, respectively (Figure 1B). Consistent with the *SlCOMT1* expression results, higher MT content was observed in the germinated seeds of WT and two overexpression lines under normal growth conditions, and the greater increment in MT content was revealed for the overexpression samples (55.44 to 88.03 pg·g^−1^ FW for OE1, 56.70 to 90.12 pg·g^−1^ FW for OE2) relative to WT (44.91 to 64.59 pg·g^−1^ FW) when being exposed to 7-d salt stress (Figure 1C).

The differences in *SlCOMT1* abundance and MT content encouraged us to explore phenotypic variations of WT and transgenic seeds in response to excessive salt supply. Under normal growth conditions, there were no significant differences in radical length between WT and two overexpression lines (Figure 1D,E). In contrast, when the seeds were exposed to 75 mM NaCl for 7 d, the growth of radicals was greatly restricted for both WT and two overexpression lines, while this inhibitory effect was apparently mitigated by *SlCOMT1* overexpression, leading to the better germination performance of stressed OE1 and OE2 seeds (Figure 1D,E). To further evaluate their growth performance, three germination-related parameters, including GP, GI and GE, were statistically analyzed for WT, OE1 and OE2 seeds. For WT samples, we observed that the GP, GI and GE values were dramatically decreased by 62.96%, 80.45% and 75% under salt stress in comparison to those under normal growing conditions, respectively (Figure 1F–H). However, the GP, GI and GE values were only reduced by 18.7%, 59.8% and 57.1% for OE1 samples, and by 8.7%, 44.8% and 48.3% for OE2 samples, upon salt stress, relative to those under normal growth conditions, respectively (Figure 1F–H). These evidences suggested that *SlCOMT1* could serve as a positive regulator in salt tolerance of tomato seeds, most likely via MT-mediated abiotic stress tolerance.

### 2.2. SlCOMT1 Overexpression Benefits Starch Metabolism in Tomato Seeds under Salt Stress

Amylase, a class of enzymes that can convert starch to soluble sugars, plays crucial roles in germination-related processes, particularly under unfavorable growth conditions, and its catalytic capacity is thus considered a key biochemical indicator of seed stress tolerance [28]. We wondered how *SlCOMT1* overexpression affected amylase activity in the stressed tomato seeds during germination process. To this end, the catalytic capacity of α-amylase, β-amylase, and total amylase was evaluated in the germinated seeds of WT and two overexpression lines under normal growth conditions and 7-d salt stress. For WT samples, we found that the activities of α-amylase, β-amylase and total amylase were largely decreased from 0.35, 3.02 and 3.37 mg·g^−1^ FW·min^−1^ under normal growth conditions to 0.14, 1.26 and 1.40 mg·g^−1^ FW·min^−1^ under 7-d salt stress, respectively (Figure 2A–C). In contrast, this inhibitory effect on amylase catalytic capacity was greatly mitigated by *SlCOMT1* overexpression: for OE1 samples, we observed that the activities of α-amylase, β-amylase, and total amylase were decreased by 36.74%, 33.30%, and 33.64% under salt stress, respectively, relative to those under normal growth conditions. For the stressed OE2 samples, 28.77%, 30.02%, and 29.89% reductions in the catalytic capacity was detected for α-amylase, β-amylase and total amylases, respectively, in comparison to those under normal growth conditions (Figure 2A–C). Consistent with the differential alterations in amylase activity, the extraordinary accumulation of starch was observed in stressed WT seeds (194.44 μg·g^−1^ FW) relative to those under normal growth conditions (98.83 μg·g^−1^ FW), while this accumulation trend was greatly decremented for the stressed samples of OE1 (163.33 μg·g^−1^ FW) and OE2 (152.81 μg·g^−1^ FW) relative to those under normal growth conditions (110.21 μg·g^−1^ FW and 113.19 μg·g^−1^ FW) (Figure 2D). These results demonstrated the beneficial effects of *SlCOMT1* overexpression on germination-related starch metabolism in stressed tomato seeds, which could at least partially account for the better germination performance of OE1 and OE2 upon salt stress.

### 2.3. SlCOMT1 Overexpression Enhances Osmotic Adjustment and Antioxidant Capacity in Tomato Seeds under Salt Stress

Upon salt stress, a series of compatible solutes, such as soluble proteins, proline, and soluble sugars, can be rapidly accumulated to adjust plant osmotic potential for water uptake, cell turgor maintenance, redox state balance, and so on, thus contributing to plant acclimation to unfavorable growth conditions [29]. To unveil whether osmotic adjustment was involved in *SlCOMT1*-mediated seed salt tolerance, we investigated the contents of soluble proteins, proline, and soluble sugars in WT, OE1 and OE2 samples. While the overaccumulation was constantly observed for soluble proteins, proline, and soluble sugars in the three stressed samples relative to those under normal growth conditions, the extents of content increase were distinct between WT (61.58% for soluble proteins, 142.73% for proline and 29.00% for soluble sugars) and the two overexpression lines (76.96%, 202.08%, and 88.58% in OE1; 73.48%, 179.55%, and 89.01% in OE2) (Figure 3A–C).

In addition to osmotic adjustment, plants have developed a highly conserved ROS scavenging system, which is mainly composed of superoxide dismutase (SOD), peroxidase (POD), and catalase (CAT), to cope with the oxidative damages caused by salt stress [30]. We further explored the responses of the antioxidant system in the germinated seeds of WT and two overexpression lines upon excessive salt supply. Similar to the abovementioned osmolyte alterations, the significant increase in SOD, POD, and CAT activities was revealed in the three stressed samples relative to those under normal growth conditions, but this stimulatory effect was much stronger in OE1 (83.72% increase for SOD, 126.62% increase for POD, and 308.27% increase for CAT) and OE2 (84.54%, 123.08%, and 333.51%) than that in WT (59.49%, 74.64%, and 145.03%) (Figure 3D–F). These observations suggested the profound involvement of both osmotic adjustment and the antioxidant system in *SlCOMT1*-mediated salt tolerance in tomato seed germination.

### 2.4. Membrane Is Stablized by SlCOMT1 Overexpression in Tomato Seeds under Salt Stress

Malonyldialdehyde (MDA) content is a critical measurement for lipid peroxidation in plants, which is commonly caused by ROS overaccumulation under unfavorable growth conditions, and can lead to a series of severe consequences such as membrane injury and physiological disorders [31]. We wondered how *SlCOMT1* overexpression influenced membrane stability in the germinated seeds under salt stress. To address this ambiguity, a microscopic investigation of ROS in general and H_2_O_2_ was first carried out in different samples using CM-H_2_DCFDA and DAB staining, respectively. The results showed that, upon 75 mM NaCl stress for 7 d, the extraordinary accumulation of ROS in general and H_2_O_2_ was observed in WT radicals, while their overaccumulation was apparently lessened in OE1 and OE2 samples (Figure 4A,B). To statistically evaluate these differences in ROS generation, H_2_O_2_ and O_2_^·−^ contents were further measured in the germinated seeds of WT and two overexpression lines. As shown in Figure 4C,D, a 693.31% increase in H_2_O_2_ content and a 176.39% increase in O_2_^·−^ content were detected in the stressed WT seeds in comparison to those under normal growth conditions. In contrast, the content increase of both biomolecules was only 369.46% and 67.95% for the stressed OE1 samples, and 195.07% and 88.67% for the stressed OE2 samples, relative to those under normal growth conditions (Figure 4C,D). Consistent with the mitigation of ROS accumulation, less enhancement of MDA content was observed in the stressed overexpression samples (96.68% for OE1 and 85.29% for OE2) than WT (164.08%), reflecting their weaker lipid peroxidation and better membrane integrity (Figure 4E).

### 2.5. SlCOMT1 Overexpression Benefits the Expression of Germination- and Tolerance-Related Genes in Tomato Seeds under Salt Stress

Seed germination, particularly under unfavorable growth conditions, is strictly controlled by ABA and GA, two phytohormones functioning as a germination inhibitor and stimulator, respectively, via imposing opposite effects on starch metabolism, etc. [18]. To provide molecular insights into *SlCOMT1*-involved seed salt tolerance, we first investigated the transcript abundance of *SlCYP707A1* (*Solyc04g078900*), *SlABA1* (*Solyc02g090890*), *SlGA3ox2* (*Solyc03g119910*), and *SlGA2ox4* (*Solyc07g061720*), four critical genes in the ABA and GA metabolic pathways [32], in different samples. For WT seeds, the expression of *SlCYP707A1*, *SlGA3ox2*, and *SlGA2ox4* was decreased by 64.80%, 84.27%, and 86.78%, respectively, while the expression of *SlABA1* was increased by 36.20% in the stressed samples relative to the control ones (Figure 5A–D). For both of the overexpression seeds, although similar trends were revealed for the abovementioned four genes, their expressional alterations were largely lessened in the stressed samples when compared to those under normal growth conditions (41.34% decrease of *SlCYP707A1*, 15.90% increase of *SlABA1*, 29.26% decrease of *SlGA3ox2*, and 38.06% decrease of *SlGA2ox4* for OE1; 33.36% decrease of *SlCYP707A1*, 16.07% increase of *SlABA1*, 29.72% decrease of *SlGA3ox2*, and 31.83% decrease of *SlGA2ox4*) (Figure 5A–D), perhaps accounting for the activation of starch metabolism by *SlCOMT1* overexpression at molecular level to some extents.

In addition to phytohormone signaling genes, a multitude of transcriptional factors (TFs) have been revealed to be profoundly involved in modulating stress tolerance in plants via stimulation of osmotic adjustment and/or the antioxidant system [33,34]. We focused on *calcium-dependent protein kinase* (*SlCDPK1*, *Solyc07g064610*), *WRKY transcription factor 33* (*SlWRKY33*, *Solyc09g014990*), and *mitogen-activated protein kinase 1* (*SlMAPK1*, *Solyc12g019460*), three representatives of salt tolerance-related TFs [35,36,37], and explored their expression in the germinated seeds of WT and the two overexpression lines under different growth conditions. The results showed that while a 0.86 fold increase for *SlCDPK1*, a 1.09 fold increase for *SlWRKY33*, and a 1.89 fold increase for *SlMAPK1* were detected in WT samples relative to those under normal growth conditions, *SlCOMT1* overexpression substantially strengthened the fold changes of *SlCDPK1* (1.60 for OE1 and 1.56 for OE2), *SlWRKY33* (2.60 for OE1 and 2.51 for OE2), and *SlMAPK1* transcript abundance (3.05 for OE1 and 3.31 for OE2) in the comparisons between NaCl and control samples (Figure 5E–G), providing some molecular explanations for *SlCOMT1*-mediated stimulation of osmotic adjustment and antioxidant capacity in the transgenic seeds upon salt challenge.

## 3. Discussion

Seed germination is a critically important stage in early plant development, which is fine-tuned by a multitude of endogenous and environmental signals [3]. MT is one of the well-known endogenous regulators in plant growth, development, and adaptability to abiotic/biotic stressors, such as seed germination under unfavorable growth conditions [15,16]. In the past several years, MT biosynthesis has been widely explored, and COMT1 is proven as a rate-limiting enzyme, of which upregulation or downregulation can promote or inhibit MT production in vivo [21,38,39,40]. In this study, we found that, in comparison to WT samples, MT content was significantly increased in *SlCOMT1* overexpression seeds under both normal and salt stress conditions, and this content enhancement was substantially strengthened upon excessive salt supply (Figure 1C). Consistent with their MT overaccumulation, the inhibitory effects of salt stress on seed germination, including radical length, GI, GP, and GE, were dramatically lessened for both overexpression samples relative to WT (Figure 1D–H), documenting the positive roles of MT produced by the *SlCOMT1* overexpression in modulating salt tolerance of tomato seeds.

Starch mobilization, a fundamental metabolic process by which insoluble carbohydrate reserves are broken down into soluble groups such as glucose and maltose, provides energy and carbon skeletons for a series of biological reactions in seed germination and early seedling development [41]. The amylolytic pathway commonly taken charge of by α-amylase and β-amylase is considered the predominant one for starch mobilization, and has been demonstrated to be positively involved in modulating seed germination performance under not only normal growth conditions but also in unfavorable environments [42]. For example, Chen et al. have reported that exogenous application of MT leads to the apparent increase of α-amylase content, together with the decrease of starch content, in cotton seeds under salt stress, thus ensuring the sufficient nutrient supply for seed germination [18]. Upon cold stress, Zhang and co-authors have observed that the catabolism of starch reserves is significantly improved in tomato seeds via the exogenous glycine betaine-mediated stimulation of α-amylase at both RNA and protein levels, benefiting the nutrient support to seed germination [43]. In the present study, we found that, in comparison to WT seeds, the activities of α-amylase, β-amylase, and total amylase were all substantially enhanced in the overexpression samples under salt stress (Figure 2A–C). Consistently, the contents of stored starch and soluble sugars were significantly decreased and increased, respectively, in the stressed seeds of two overexpression lines relative to WT samples (Figure 2D and Figure 3C). These observations pointed to the significance of *SlCOMT1* in starch mobilization during tomato seed germination, which could provide not only the nutrient sources for the germination-related processes, but also the supportive substances for osmotic adjustment, thus greatly contributing to seed salt tolerance and germination performance upon excessive salt supply by elevating MT production.

Excessive salt can impose osmotic stress on seeds to disturb their capacity of water absorption, which is a prerequisite for a series of programmed biochemical events during seed germination, including the expression of germination-related genes [32], the activation of various enzymatic systems [44], the mobilization of storage substance reserves [45], and so on. The delaying and/or prevention of seed germination have been widely reported for field and horticultural crops, such as cotton [18], rice [46], banana [47] and wheat [48], upon salt challenge. To cope with a series of stressful conditions, plants have evolved the rapid osmotic adjustment mechanism, a biochemical process by which a multitude of osmotic regulatory substances can be dramatically accumulated to improve the water retention capacity of cells and maintain the integrity of cell membranes [29]. In this study, the overaccumulation of soluble proteins, proline, and soluble sugars, three key osmotic regulatory substances, was observed in the germinated seeds of two overexpression lines relative to WT samples when being challenged by 75 mM NaCl (Figure 3A–C). Similarly, in a study carried out by Castañares and Bouzob, the germination of melon seeds has been promoted under salt stress by exogenous application of MT, which imposes beneficial effects on the accumulation of osmotic regulatory substances, and thereby improves the capacity of osmotic adaption [49]. Chen et al. have also reported that the inhibition of cotton seed germination is efficiently mitigated by MT pretreatment, mainly due to the promoted contents of soluble proteins, proline, and soluble sugars under salt stress [50]. All abovementioned evidences thus revealed the conservative roles of MT, the production of which is crucially catalyzed by *COMT1* [20,21] in the acclimation of germinated seeds to salt stress via modulating osmotic adjustment.

In addition to the restricted water availability, excessive salt can induce the extraordinary generation of ROS, such as H_2_O_2_ and O_2_^·−^, in stressed seeds, which triggers oxidative stress to cause great damage to plasma membraned and critical biochemical events such as lipid peroxidation [31]. To eliminate these injuries by oxidative stress, a multitude of enzymatic antioxidants, including SOD, POD, CAT, ascorbate oxidase, glutathione peroxidase, and glutathione reductase, constitute a sophisticated scavenging system of ROS in plants [30]. A number of studies have demonstrated the incremented activities of antioxidant enzymes in stressed seeds, and their catalytic capacity is thus proposed to be an important measure for seed salt tolerance [51,52,53]. In the present study, although the activities of SOD, POD, and CAT were significantly increased upon salt stress in all stressed tomato seeds, the extents of activity enhancement were distinct between WT and overexpression samples: for WT, 59.49%, 74.64%, and 145.03% increases were detected in the activities of SOD, POD, and CAT, respectively; while for OE1 and OE2, 83.72% and 84.54% increases in SOD activity, 126.62% and 123.08% increases in POD activity, and 308.27% and 333.51% increases in CAT activity were observed, respectively (Figure 3D–F). Consistent with the higher antioxidant capacity, we found that, under salt stress, the burst of H_2_O_2_ and O_2_·^−^ was dramatically mitigated in the germinated seeds of two overexpression lines in comparison to WT samples, leading to lowered MDA accumulation (Figure 4). Similar results have been reported in the germinated seeds of cotton [54], banana [47], and wheat [48] under salt stress. Given the fact that MT, either endogenous or exogenous, is profoundly involved in ROS scavenging [55,56,57], we assumed that *SlCOMT1* overexpression, which stimulated endogenous MT production, might modulate oxidative stress and ROS generation in the geminated seeds through the enzymatic antioxidant system, thereby enhancing their tolerance to salt stress.

Recently, a close association of MT-mediated seed germination has been revealed with the metabolic activities of ABA and GA, which commonly exhibit inhibitory and stimulatory effects on seed germination, respectively [18]. For example, Zhang et al. have reported that the inhibitory effects of salt stress on cucumber seed germination can been apparently mitigated by MT pretreatment, mainly due to the enhancement in both ABA degradation and GA biosynthesis [58]. In another study carried out by Li et al., the authors revealed an antagonistic interaction existed between MT and ABA, while a stimulatory interaction existed between MT and GA, leading to the significantly decreased ABA/GA ratio and the stimulated germination of melon seeds under stressful conditions [59]. In the present study, we investigated the expressions of *SlCYP707A1* and *SlABA1*, two key genes responsible for ABA biosynthesis and catabolism, respectively, and *SlGA3ox2* and *SlGA2ox4*, two critical components in the GA anabolic and catabolic pathways, respectively [32], in different tomato samples. For WT seeds, 64.80%, 84.27%, and 86.78% decreases of transcript abundance were observed for *SlCYP707A1*, *SlGA3ox2*, and *SlGA2ox4*, respectively, while a 36.20% increase was detected for *SlABA1*, in the stressed samples relative to the control ones. In contrast, for OE1 and OE2 seeds, although the similar expressional trends were detected for the abovementioned four genes, their abundance alterations were greatly lessened in the stressed samples relative to those under normal growth conditions (41.34% decrease of *SlCYP707A1*, 15.90% increase of *SlABA1*, 29.26% decrease of *SlGA3ox2*, and 38.06% decrease of *SlGA2ox4* for OE1; 33.36% decrease of *SlCYP707A1*, 16.07% increase of *SlABA1*, 29.72% decrease of *SlGA3ox2*, and 31.83% decrease of *SlGA2ox4* for OE2) (Figure 5A–D). These results supported the assumption that *SlCOMT1* overexpression, which stimulated endogenous MT production, could mitigate the inhibitory effects of salt stress on seed germination via reconstituting ABA and GA metabolism to some extents.

A number of TFs have been shown to be profoundly involved in modulating plant stress responses [47,60]. For example, the positive roles of *ClCDPK* and *ClWRKY* in MT-mediated tolerance for watermelon seedlings under cold stress [60], and of *SlCDPK1* and *SlMAPK1* in MT-mediated tolerance for tomato seedlings under multiple abiotic stresses, have been observed in the studies carried out by Li et al. [60] and Gong et al. [35], respectively. More recently, in a study carried out by Wei et al., they found that *MnWRKY* might function as one of the key positive regulators for MT-mediated acclimation of banana seedlings under salt stress with the aid of transcriptomic sequencing [47]. In this study, a 0.86 fold increase for *SlCDPK1*, a 1.09 fold increase for *SlWRKY33*, and a 1.89 fold increase for *SlMAPK1* were observed in stressed WT seeds in comparison to those under normal growth conditions, whereas their upregulation was dramatically strengthened for both overexpression samples (1.60 fold increase for *SlCDPK1*, 2.60 fold increase for *SlWRKY33*, and 3.05 fold increase for *SlMAPK1* in OE1; 1.56 fold increase for *SlCDPK1*, 2.51 fold increase for *SlWRKY33*, and 3.31 fold increase for *SlMAPK1* in OE2) (Figure 5E–G), providing some molecular explanations for endogenous MT-mediated stimulation of osmotic adjustment and the antioxidant system in the overexpression seeds upon salt challenge.

Based on the observations in this study, we proposed a working model to explain *SlCOMT1*-involved improvement of seed salt tolerance (Figure 6): upon salt stress, *SlCOMT1* overexpression induced the overaccumulation of MT in tomato seeds, which, on one hand, stimulated the mobilization of food reserves, such as starch, to ensure nutrient requirements via reconstituting ABA and GA metabolism, and, on the other hand, improved the capacity of both osmotic adjustment and ROS scavenging through activating tolerance-related signaling to prevent stress-derived damages. As a result, enhanced salt tolerance and better germination performance could be obtained for tomato seeds.

## 4. Materials and Methods

### 4.1. Plant Materials and Experimental Designs

Tomato (*Solanum lycopersicum* L.) inbred line ‘895’ (wild type, WT) and two *SlCOMT1* overexpression lines (OE1 and OE2), which were obtained via the agrobacterium-mediated transformation under the WT genetic background [26,27], were used in this study. The sterilized seeds were subjected to an 8 h water bath followed by a 16-h incubation at 28 °C in darkness. Thereafter, the seeds were sandwiched with Whatman filter paper in bacteria-free petri dishes, and salt treatment was immediately carried out according to the following experimental designs: (I) Control, wherein the Whatman paper was wetted by 5 mL of sterilized water; and (II) NaCl, wherein the Whatman paper was wetted by 5 mL of 75 mM NaCl solution. The stressed seeds were placed in an incubator without lighting at 28 °C. At 7 days after treatment, the Control and NaCl seeds were collected for subsequent assay. Three biological repeats were performed for each parameter.

### 4.2. Evaluation of Seed Germination

Seed germination-related parameters, including germination percentage (GP), germination index (GI), and germination energy (GE), were calculated according to the following formula: (I) GP (%) = (n/N) × 100, wherein ‘n’ and ‘N’ represent the number of germinated seeds and the total number of seeds used in the assay, respectively [61]; (II) GI = ∑(G/t), wherein ‘G’ and ‘t’ represent the number of germinated seeds at a specific investigation time and the total time of germination period respectively [62]; and (III) GE = N_1_/D_1_ + (N_2_ − N_1_)/D_2_ + ⋯ + (N_j_ − N_i_)/D_j_, wherein ‘N’ and ‘D’ represent the number of germinated seeds at the specific investigation day and the total days of germination period, respectively [61].

### 4.3. Determination of MT Content

MT content was determined with the Plant MT ELISA Kit (MLBIO, Shanghai, China) according to the manufacturer’s instructions. In brief, the mixture of 10 μL MT extracts and 100 reaction buffer was incubated in a sealed reaction well at 37 °C for 60 min. After being washed five times, a 50 μL chromogenic agent A and a 50 μL chromogenic agent B were added to the reaction well, mixed by gently shaking, and subjected to a 15 min incubation at 37 °C in darkness. Finally, 50 μL of termination buffer was added to the reaction well, and the absorbance at 450 nm was monitored for the calculation of MT content according to the standard curve, which was parallelly prepared.

### 4.4. Starch Metabolism Assay

The activities of α- amylase, β-amylase, and total amylase were determined according to the previously described methods [46,63] with minor modifications. In brief, 1 mL enzymatic extract was first mixed well with 1 mL of 1% (*w*/*v*) soluble starch, which was prepared in sodium acetate buffer (pH 5.6), and then subjected to a 15 min incubation at 40 °C. After the addition of 2 mL 3,5-dinitrosalicylic acid, the resulting mixture was boiled for 5 min and spectrometrically monitored at 540 nm for the calculation of total amylase activity. α-amylase activity was determined by following the abovementioned protocol for total amylase, except a 15 min heating treatment at 70 °C. β-amylase activity was finally deduced from the difference between total amylase and α-amylase activities. For the starch assay, 0.3 g seed samples were well ground and subjected to two-round extraction with 80% (*v*/*v*) ethanol followed by 52% (*v*/*v*) perchloric acid. The starch content was then determined according to the anthrone colorimetry method [64]. To analyze soluble sugars, the homogenized seed samples were first incubated at 80 °C for 30 min. After centrifugation, 0.5 mL of the resulting extracts were mixed with 1.5 mL ddH_2_O and 0.5 mL anthrone-ethyl acetate solution for another incubation at 80 °C for 15 min. The absorbance at 620 nm was measured to calculate soluble sugar content by following the previously described protocol [64]. For soluble protein assay, 0.3 g seed samples were well ground with 5 mL 50 mM phosphate buffer solution (PBS, pH 7.8), transferred to a 50 mL centrifuge tube, and then subjected to 20 min centrifugation with 10,000× *g* at 4 °C. The resulting supernatant was used to determine soluble protein content using the coomassie brilliant blue G-250 method [47].

### 4.5. Evaluation of Antioxidant Enzyme Activity

SOD activity was determined through monitoring its inhibitory capacity for the photochemical reduction of nitro-blue tetrazolium according to the method described by Stewart and Bewley [65]. POD activity was assayed by following the guaiacol oxidation-based protocol, wherein the absorbance variations of the reaction mixture was detected at 470 nm for evaluating the enzymatic capacity of catalyzing oxidation-reduction reaction [66]. For CAT, the absorbance decline at 240 nm was detected in the reaction mixture, which was composed of 25 mM phosphate buffer (pH 7.0), 10 mM H_2_O_2_, and enzyme extract, and used for assessing its decomposition capacity of H_2_O_2_ by following the Patra et al.’s protocol [67].

### 4.6. Determinaiton of ROS, MDA and Proline Contents

ROS visualization was performed by following the previously described protocols [68] with minor modifications. To localize ROS generation, tomato radicals were stained with CM-H_2_DCFDA (C6827, Invitrogen, Waltham, MA, USA), a general oxidative stress indicator, and then examined under a laser scanning confocal microscope (LSM-510 Meta, Zeiss, Shanghai, China) with the excitation wavelength of 488 nm and the emission wavelength of 500–530 nm. Histochemical staining analysis for H_2_O_2_ was carried out with 3,3′-diaminobenzidine (DAB) by following the previously described protocol [69]. To further quantify H_2_O_2_, the homogenized seed samples was centrifuged with 10,000× *g* at 4 °C for 20 min. A volume of 1 mL of the resulting supernatant was then mixed well with 1 mL extraction buffer (CCl_4_:CH_3_Cl_3_ = 3:1, *v*/*v*) and 3 mL of ddH_2_O, and subjected to another centrifugation with 6000× *g* at 4 °C for 10 min. A volume of 1 mL of the top layer solution was reacted with 0.1 mL 20% (*w*/*v*) TiSO4 solution and 0.2 mL concentrated ammonia solution, and the pellet was dissolved in 5 mL 2 M H_2_SO_4_ solution for the determination of H_2_O_2_ content by following the previously describe protocol by Yan et al. [26]. For O_2_^·−^ assay, 0.3 g seed samples were well ground by adding 5 mL 65 mM PBS (pH 8.0), and the homogenate was then centrifuged with 10,000× *g* at RT for 15 min. A volume of 0.5 mL of the resulting supernatant was mixed with 1 mL hydroxylamine and incubated at 25 °C for 60 min. After adding 1 mL 17 mM p-aminobenzene sulfonic acid and 1 mL 7 mM α-naphthylamine solution, the reaction mixture was subjected to another incubation at 25 °C for 20 min, and the absorbance at 530 nm was measured to calculate O_2_^·−^ productivity rate according to the method described before [26].

To monitor MDA production, 0.2 g seed samples were homogenized in 3 mL of 10% (*w*/*v*) trichloroacetic acid (TCA) with a pestle and mortar, and subjected to a 10 min centrifugation with 10,000× *g* at 4 °C. A 1 mL aliquot from the supernatant was well mixed with the same volume of 0.5% (*w*/*v*) thiobarbituric acid (TBA), which was prepared in 10% (*v*/*v*) TCA solution, and incubated at 95 °C for 30 min. After another 10 min centrifugation with 10,000× *g* at 28 °C, the absorbance of resulting supernatant was detected at 532 nm and 600 nm, respectively, and MDA content was calculated by following the method described by Wang et al. [70]. For the determination of proline content, seed samples were homogenized in 3% (*w*/*v*) sulfo-salicylic acid and centrifuged at 6000× *g* for 10 min. The resulting supernatant together with ninhydrin and glacial acetic acid was heated at 100 °C for 60 min. The reaction mixture was further extracted with 4 mL toluene by vigorously vortexing for 30 s, and the absorbance was measured at 520 nm to calculate proline content according to the previously described method [71].

### 4.7. Gene Expression Assay

Total RNAs were extracted with the TRIzol reagent (Cat No. 15596018, Invitrogen, USA) according to the manufacturer’s instructions. The first strand complementary DNAs (cDNAs) were then synthesized using the Revert Aid First Strand cDNA Synthesis Kit (R233-01, Vazyme, Nanjing, China), and quantitative PCR (qPCR) was carried out with the Power SYBR Green PCR Master Mix (Cat No. 4367659, ABI, Life Tech, Carlsbad, CA, USA) on an ABI 7900 HT Fast Real-Time PCR System. For each gene of interest, three biological and technical repeats were performed, respectively, and the relative expression level was calculated according to the 2^−∆∆CT^ algorithm [72]. The primers used for these quantitative assays are provided in Table 1.

### 4.8. Statistical Analysis

All experimental data were statistically processed with the SPSS v10.0 software (SPSS Inc., Chicago, IL, USA) by following the multiple comparison rules of Tukey’s test, and further visualized with the GraphPad Prism v8.0 software (GraphPad Software Inc., San Diego, CA, USA).

## 5. Conclusions

*SlCOMT1* overexpression stimulated the endogenous MT signaling in tomato seeds, which could confer the high acclimation capacity to environmental stressors, and the better germination performance was thus observed under salt stress. Physiological investigation revealed that the *SlCOMT1*-mediated germination improvement was at least partially attributed to the promotion of starch metabolism, as well as the enhancement of osmotic adjustment and antioxidant capacity in the stressed overexpression seeds. The distinct alterations in transcript abundance of hub genes related to germination (*SlCYP707A1*, *SlABA1*, *SlGA3ox2*, and *SlGA2ox4*) and stress tolerance (*SlCDPK1*, *SlWRKY33*, and *SlMAPK1*) were observed in the overexpression seeds in comparison to WT samples upon salt stress, providing some molecular explanations for the abovementioned physiological observations. Taken together, our study sheds new molecular-physiological insights into *SlCOMT1*-mediated salt tolerance of tomato seeds and could contribute to the MT-based genetic engineering of crops in future.

## Figures and Tables

**Figure 1 ijms-24-00734-f001:**
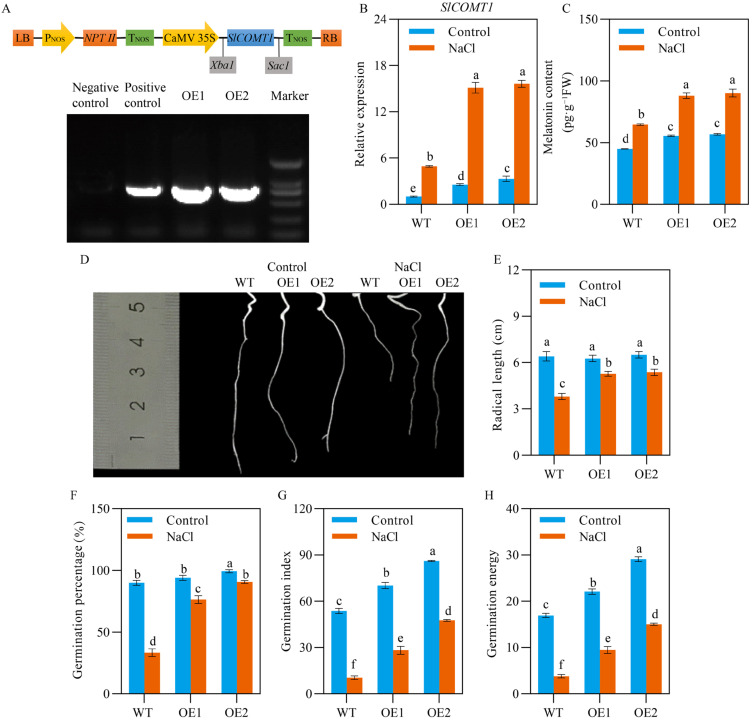
Effects of *SlCOMT1* overexpression on tomato seed germination under salt stress. (**A**) Schematic illustration for T-DNA region of the binary vector pROKII-*SlCOMT1* used for tomato transformation (top panel), and genotyping analysis for the germinated seeds of two overexpression lines (OE1 and OE2) used in this study (bottom panel). LB and RB: left and right boards; P_NOS_ and T_NOS_: *NOS* promoter and terminator; CaMV 35S: CaMV 35S promoter; *NPTII*: kanamycin resistance gene; *SlCOMT1*: tomato *caffeic acid O-methyltransferase 1*. (**B**,**C**) Relative expression of *SlCOMT1* (**B**) and melatonin (MT) content (**C**) in the germinated seeds of wild type (WT) and two overexpression lines (OE1 and OE2) under normal growth conditions (Control) and 7-d salt stress (NaCl). (**D**,**E**) Phenotypic (**D**) and quantitative analysis (**E**) for the radicals of wild type (WT) and two overexpression lines (OE1 and OE2) under normal growth conditions (Control) and 7-d salt stress (NaCl). (**F**–**H**) Quantitative analysis for three germination-related parameters, including germination potential (GP, **F**), germination index (GI, **G**) and germination energy (GE, **H**), in the seeds of wild type (WT) and two overexpression lines (OE1 and OE2) under normal growth conditions (Control) and 7-d salt stress (NaCl). In (**B**,**C**,**E**–**H**), each parameter displays as mean of three biological repeats ± standard deviation, and different letters indicate significant differences between WT, OE1 and OE2 samples under normal growth conditions (Control) and 7-d salt stress (NaCl) by following the multiple comparison rules of Tukey’s test (*p* < 0.05).

**Figure 2 ijms-24-00734-f002:**
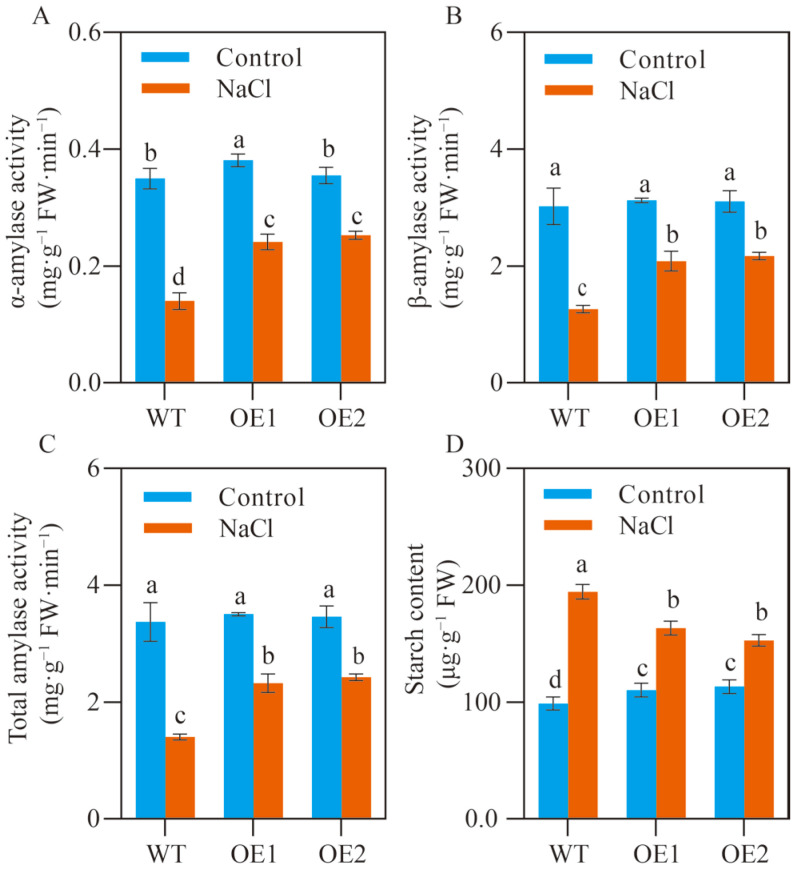
Effects of *SlCOMT1* overexpression on starch metabolism in tomato seeds under salt stress. (**A**–**C**) α-amylase (**A**), β-amylase (**B**) and total amylase activities (**C**) in the germinated seeds of wild type (WT) and two overexpression lines (OE1 and OE2) under normal growth conditions (Control) and 7-d salt stress (NaCl). (**D**) Starch content in the germinated seeds of wild type (WT) and two overexpression lines (OE1 and OE2) under normal growth conditions (Control) and 7-d salt stress (NaCl). Each parameter displays as mean of three biological repeats ± standard deviation, and different letters indicate significant differences between WT, OE1 and OE2 samples under normal growth conditions (Control) and 7-d salt stress (NaCl) by following the multiple comparison rules of Tukey’s test (*p* < 0.05).

**Figure 3 ijms-24-00734-f003:**
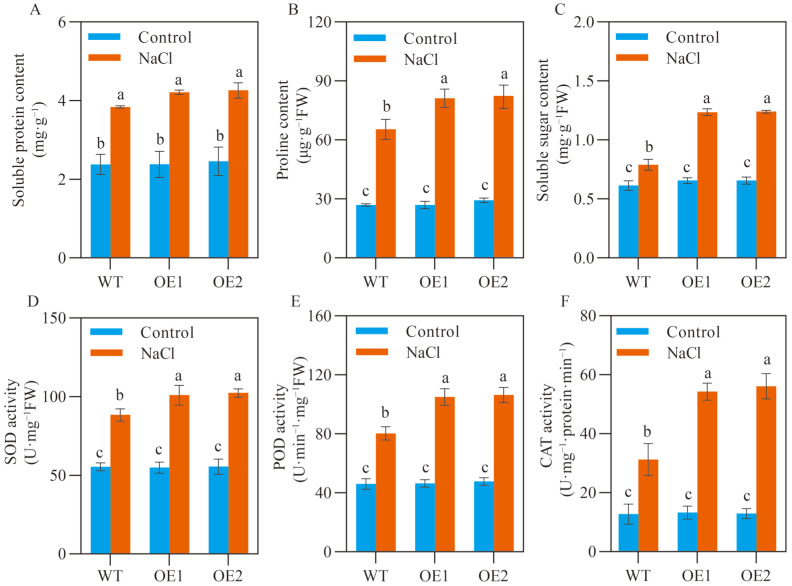
Effects of *SlCOMT1* overexpression on osmolyte accumulation and antioxidant enzymatic activity in tomato seeds under salt stress. (**A**–**C**) Soluble protein (**A**), proline (**B**) and soluble sugar contents (**C**) in the germinated seeds of wild type (WT) and two overexpression lines (OE1 and OE2) under normal growth conditions (Control) and 7-d salt stress (NaCl). (**D**–**F**) SOD (**D**), POD (**E**) and CAT activities (**F**) in the germinated seeds of wild type (WT) and two overexpression lines (OE1 and OE2) under normal growth conditions (Control) and 7-d salt stress (NaCl). Each parameter displays as mean of three biological repeats ± standard deviation, and different letters indicate significant differences between WT, OE1 and OE2 samples under normal growth conditions (Control) and 7-d salt stress (NaCl) by following the multiple comparison rules of Tukey’s test (*p* < 0.05).

**Figure 4 ijms-24-00734-f004:**
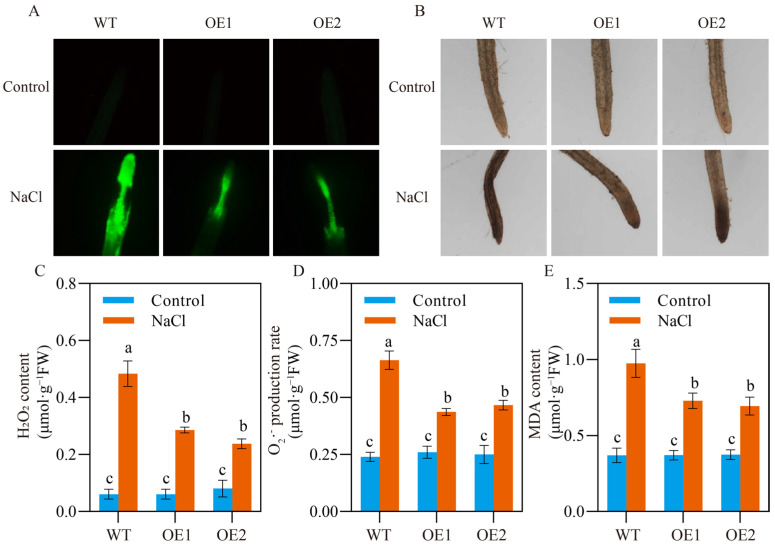
Effects of *SlCOMT1* overexpression on ROS and MDA production in tomato seeds under salt stress. (**A**,**B**) ROS (**A**) and H_2_O_2_ generation (**B**) in the germinated seeds of wild type (WT) and two overexpression lines (OE1 and OE2) under normal growth conditions (Control) and 7-d salt stress (NaCl). (**C**–**E**) H_2_O_2_ (**C**), O_2_^·−^ (D) and MDA contents (**E**) in the germinated seeds of wild type (WT) and two overexpression lines (OE1 and OE2) under normal growth conditions (Control) and 7-d salt stress (NaCl). Each parameter displays as mean of three biological repeats ± standard deviation, and different letters indicate significant differences between WT, OE1 and OE2 samples under normal growth conditions (Control) and 7-d salt stress (NaCl) by following the multiple comparison rules of Tukey’s test (*p* < 0.05).

**Figure 5 ijms-24-00734-f005:**
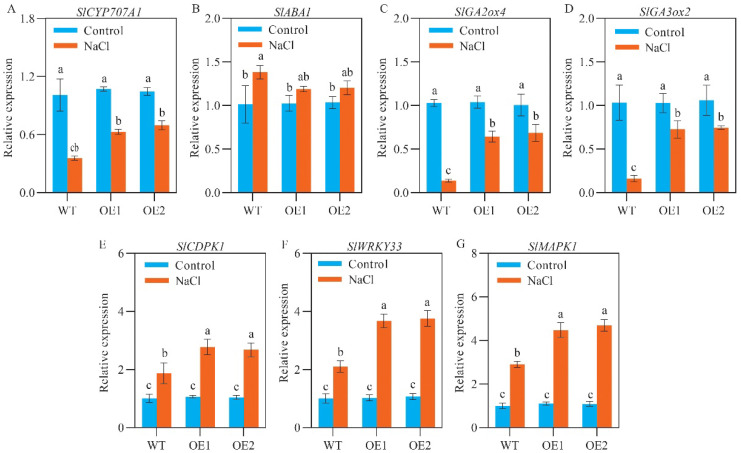
Effects of *SlCOMT1* overexpression on the abundance of germination- and tolerance-related genes in tomato seeds under salt stress. (**A**,**B**) Relative expression of *SlCYP707A1* (**A**) and *SlABA1* (**B**), two ABA metabolic genes, in the germinated seeds of wild type (WT) and two overexpression lines (OE1 and OE2) under normal growth conditions (Control) and 7-d salt stress (NaCl). (**C**,**D**) Relative expression of *SlGA2ox4* (**C**) and *SlGA3ox2* (**D**), two representatives of GA metabolic genes, in the germinated seeds of wild type (WT) and two overexpression lines (OE1 and OE2) under normal growth conditions (Control) and 7-d salt stress (NaCl). (**E**–**G**) Relative expression of *SlCDPK1* (**E**), *SlWRKY33* (**F**) and *SlMAPK1* (**G**), three representatives of stress tolerance-related transcriptional factors, in the germinated seeds of wild type (WT) and two overexpression lines (OE1 and OE2) under normal growth conditions (Control) and 7-d salt stress (NaCl). Each parameter displays as mean of three biological repeats ± standard deviation, and different letters indicate significant differences between WT, OE1 and OE2 samples under normal growth conditions (Control) and 7-d salt stress (NaCl) by following the multiple comparison rules of Tukey’s test (*p* < 0.05).

**Figure 6 ijms-24-00734-f006:**
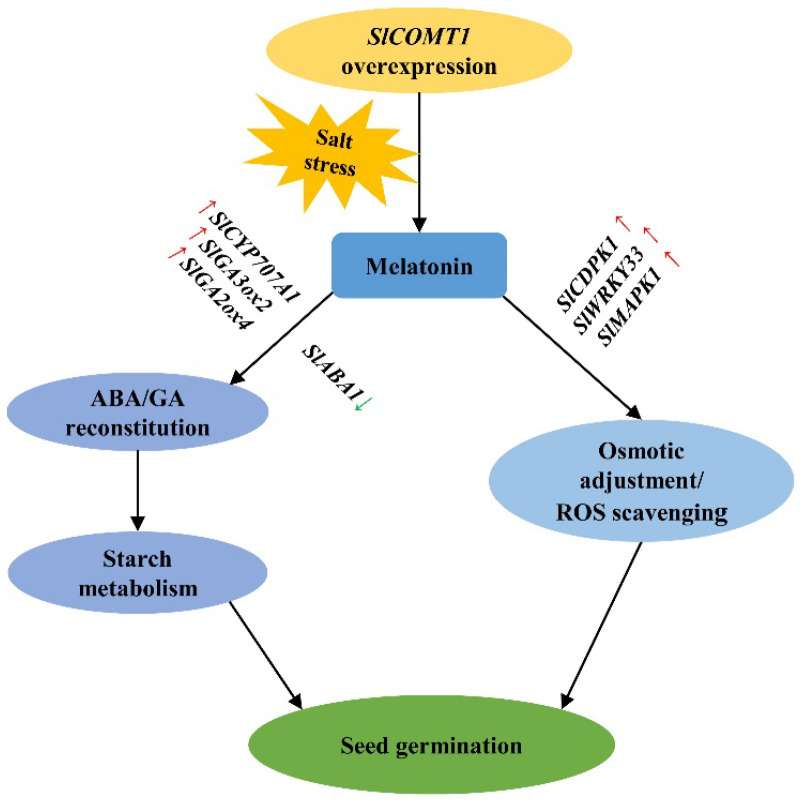
A working model to explain *SlCOMT1*-involved improvement of seed salt tolerance. *SlCOMT1* overexpression induced the overaccumulation of MT in tomato seeds upon salt stress, which stimulated the mobilization of food reserves such as starch to ensure nutrient requirements via reconstituting ABA and GA metabolism, and, simultaneously, improved the capacity of both osmotic adjustment and ROS scavenging through activating tolerance-related signaling to prevent stress-related damages. As a result, the salt tolerance of tomato seeds overexpressing *SlCOMT1* was dramatically improved, thereby displaying the better germination performance. The red and green arrows represent the up- and down-regulated genes by MT, respectively.

**Table 1 ijms-24-00734-t001:** The primers used for genotyping and qPCR assays.

Gene Name	Forward Primer	Reverse Primer
*Actin*	TTTGCTGGTGATGATGCC	CCTTAGGGTTGAGAGGTGCTT
*NPTII*	GTGGAGAGGCTATTCGGCTATGACTG	AGCTCTTCAGCAATATCACGGGTAGC
*SlCOMT1*	TACCCTGGCGTTGAACACA	CCTTTCTTTGCCTCCTGGATTA
*SlCYP707A1*	ATCACAACCCAGAGTTCTTTCCT	CAAGTTCATTCCCTGGACAAGC
*SlGA3ox2*	GATAAGCTCATGTGGTCCGAAG	GCTTTTCCATTTCATTTTCGTA
*SlGA2ox4*	TGGCAATAAGAAAATCGGACAA	ACACATAATCATTCACCGCAGC
*SlABA1*	AGAGTCTGGAAGCCCTGTGGAT	AAGTCCGACGCCAAGATAAGC
*SlMAPK1*	GGTGGCAGGTTCATTCAATAC	TTCTCTCTGTGGTGGTGGAA
*SlWRKY33*	CTACAGTGTTGGCTAACCATTCTAAT	GTTAAGGAAAGAGCTGAAGAATAAATCA
*SlCDPK1*	TCTTGTGATGGAGTTGTGTGG	AATGAATACCGACAGCCCA

## Data Availability

Not applicable.

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
