# Peer review of "Improvement of Seed Germination under Salt Stress via Overexpressing Caffeic Acid O-methyltransferase 1 (SlCOMT1) in Solanum lycopersicum L."

_ijms, 2023, doi:10.3390/ijms24010734_

Round 1
Reviewer 1 Report
Manuscript entitled " Improvement of seed germination under salt stress via overexpressing cafeic acid O-methyltransferase 1 (SlCOMT1) in Solanum lycopersicum L" submitted by Shi deals with the study of the effect of melatonin/SlCOMT1 on tomato seed germination under salt stress. The outcomes of this study are quite clearly presented in the MS. However, revision is necessary to make this paper more presentable for the readers. Following are my minor suggestions which will help authors to improve their work.
1. Is this study focused on COMT1 or MT effects on seeds? In my opinion, the role of MT should be emphasized, not just COMT1. So, more description to MT should be included in the Introduction and Discussion.
2. In the results, I think should add the results of the effect of exogenous MT on seeds under salt stress.
3. In the Introduction, the Authors are suggested to write more detail about the effect of salt stress on seed germination.
4. In the Introduction, more content about COMT should be added.
5. Line 364-366: further explanation ‘All evidence thus revealed the conservative roles of SlCOMT1-mediated osmotic adjustment in the acclimation of germinated seeds to salt stress.’
6. Please add a description of the role of MT on ROS metabolism in the Discussion. There are some related articles that can help the author e.g.
Yu et al., Journal of Pineal Research, 2022, e12810;
Sun et al., Journal of Integrative Plant Biology, 2021, 63(1), 126-145
Zeng et al., Frontiers in plant science, 2022, 13. 847175
7. In Materials and Methods. The seeds were sandwiched with Whatman paper and wetted by sterilized water or NaCl solution, and then treated for 7 days. Did you keep the Whatman paper moist during treatment processing or did you only add water/NaCl once?
8. I suggest that the authors rewrite the ROS determination section in 4.6 more clearly.
9. In Author Contributions: SS does not appear in the authors.
Author Response
Dear Reviewers,
Thank you very much for reviewing our manuscript “Improvement of seed germination under salt stress via overexpressing caffeic acid O-methyltransferase 1 (SlCOMT1) in Solanum lycopersicum L. (ijms-2106446)” and providing insightful comments and invaluable suggestions. Based on your comments and suggestions, we have revised our manuscript and prepared the point-by-point responses to your comments below.
Point-by-point responses to Reviewer 1’s comments
Comments and Suggestions for Authors
Manuscript entitled "Improvement of seed germination under salt stress via overexpressing cafeic acid O-methyltransferase 1 (SlCOMT1) in Solanum lycopersicum L" submitted by Shi deals with the study of the effect of melatonin/SlCOMT1 on tomato seed germination under salt stress. The outcomes of this study are quite clearly presented in the MS. However, revision is necessary to make this paper more presentable for the readers. Following are my minor suggestions which will help authors to improve their work.
(1) Is this study focused on COMT1 or MT effects on seeds? In my opinion, the role of MT should be emphasized, not just COMT1. So, more description to MT should be included in the Introduction and Discussion.
Response: Thank you very much for your comments. Factually, the objective of this manuscript is to elucidate how endogenous MT influences seed germination in response to salt stress by overexpressing the SlCOMT1, which encodes a rate-limiting enzyme in MT biosynthesis. Accordingly, we added more descriptive information about MT in both Introduction and Discussion sections to emphasize the important roles of MT in plants.
(2) In the results, I think should add the results of the effect of exogenous MT on seeds under salt stress.
Response: Thank you very much for your invaluable suggestions. We completely agreed to that more supportive information can be provided for our conclusions if an exogenous MT treatment would be carried out to investigate seed germination under salt stress. We were very glad to performed this experiment, but unfortunately the wet work in SDAU is now halted due to the severe Covid-19 pandemic. Given the 5-d revision period, we are afraid that this recommended experiment could not be carried out. Anyway, this point will be seriously considered in our subsequent studies.
(3) In the Introduction, the Authors are suggested to write more detail about the effect of salt stress on seed germination.
Response: The more detailed information about the effect of salt stress on seed germination was added in the section of “Introduction”.
(4) In the Introduction, more content about COMT should be added
Response: The more detailed information about COMT was added in the section of “Introduction”.
(5) Line 364-366: further explanation ‘All evidence thus revealed the conservative roles of SlCOMT1-mediated osmotic adjustment in the acclimation of germinated seeds to salt stress.’
Response: This was done.
(6) Please add a description of the role of MT on ROS metabolism in the Discussion. There are some related articles that can help the author e.g.
Yu et al., Journal of Pineal Research, 2022, e12810;
Sun et al., Journal of Integrative Plant Biology, 2021, 63(1), 126-145
Zeng et al., Frontiers in plant science, 2022, 13. 847175
Response: The more description about the role of MT on ROS metabolism was added in the section of “Discussion” according to the recommended paper, which were cited in the revised manuscript as well.
(7) In Materials and Methods. The seeds were sandwiched with Whatman paper and wetted by sterilized water or NaCl solution, and then treated for 7 days. Did you keep the Whatman paper moist during treatment processing or did you only add water/NaCl once?
Response: The Whatman paper used to sandwich tomato seeds was kept moist during over the 7-day treatment period by adding the equal volume of water or NaCl solution.
(8) I suggest that the authors rewrite the ROS determination section in 4.6 more clearly.
Response: This was done.
(9) In Author Contributions: SS does not appear in the authors.
Response: This revision was done

Reviewer 2 Report
The article “Improvement of seed germination under salt stress via overexpressing caffeic acid O-methyltransferase 1 (SlCOMT1) in Solanum lycopersicum” is very interesting. It has a lot of results. It is concisely written. All results are pretty clear and we can see a statistically significant difference between control and treatment under salt stress. Nice results.
I have a few comments:
1. Please correct the word “cafeic” into “caffeic” in a whole article.
Line 3, line 17, line 145, line 621
2. Please correct the word “derivation” into “deviation”.
Line 189, line 225, line 304
3. You could check how it is correct to write statistical level at the end of figures legend: 0.05 or p≤0.05 or somehow different.
4. In the section Materials and Methods, it’s missing how you did the analysis for soluble protein content, prolin content and soluble sugar content.
Author Response
Dear Reviewers,
Thank you very much for reviewing our manuscript “Improvement of seed germination under salt stress via overexpressing caffeic acid O-methyltransferase 1 (SlCOMT1) in Solanum lycopersicum L. (ijms-2106446)” and providing insightful comments and invaluable suggestions. Based on your comments and suggestions, we have revised our manuscript and prepared the point-by-point responses to your comments below.
Point-by-point responses to Reviewer 2’s comments
Comments and Suggestions for Authors
The article “Improvement of seed germination under salt stress via overexpressing caffeic acid O-methyltransferase 1 (SlCOMT1) in Solanum lycopersicum” is very interesting. It has a lot of results. It is concisely written. All results are pretty clear and we can see a statistically significant difference between control and treatment under salt stress. Nice results.
I have a few comments:
(1) Please correct the word “cafeic” into “caffeic” in a whole article.
Line 3, line 17, line 145, line 621
Response: The revision was done.
(2) Please correct the word “derivation” into “deviation”.
Line 189, line 225, line 304
Response: This was done.
(3) You could check how it is correct to write statistical level at the end of figures legend: 0.05 or p≤0.05 or somehow different.
Response: This revision was done.
(4) In the section Materials and Methods, it’s missing how you did the analysis for soluble protein content, prolin content and soluble sugar content.
Response: The detailed information about the analysis for soluble protein, proline content and soluble sugar contents was added in the section of “Materials and Methods”.
